# Assessment of 3D Lumbosacral Vascular Anatomy for OLIF51 by Non-Enhanced MRI and CT Medical Image Fusion Technique

**DOI:** 10.3390/diagnostics11101744

**Published:** 2021-09-23

**Authors:** Masakazu Nagamatsu, Sameer Ruparel, Masato Tanaka, Yoshihiro Fujiwara, Koji Uotani, Shinya Arataki, Taro Yamauchi, Yoshiyuki Takeshita, Rika Takamoto, Masato Tanaka, Shinsuke Moriue

**Affiliations:** Department of Radiology, Okayama Rosai Hospital, 1-10-25 Chikkomidorimachi, Minami Ward Okayama, Okayama 702-8055, Japan; n.masa1987215@gmail.com (M.N.); ruparelsameer@gmail.com (S.R.); fujiwarayoshihiro2004@yahoo.co.jp (Y.F.); coji.uo@gmail.com (K.U.); araoyc@gmail.com (S.A.); ygitaro0307@yahoo.co.jp (T.Y.); takerad@gmail.com (Y.T.); naskal.rika.15@gmail.com (R.T.); rsm19811102@gmail.com (S.M.)

**Keywords:** OLIF51, vascular anatomy, multi-modality image fusion, medical image

## Abstract

Study design: Prospective study. Objective: Medical image fusion can provide information from multiple modalities in a single image. The present study aimed to determine whether three-dimensional (3D) lumbosacral vascular anatomy could be adequately portrayed using a non-enhanced CT–MRI medical image fusion technique. Summary of Background Data: Lateral lumbar interbody fusion has gained popularity for the surgical treatment of adult spinal deformity (ASD). Oblique lumbar interbody fusion at L5–S1 (OLIF51) is receiving considerable attention as a method of creating good L5–S1 lordosis. Access in OLIF51 requires evaluation of the vascular anatomy in the lumbosacral region. Conventional imaging modalities need a contrast medium to describe the vascular anatomy. Methods: Participants comprised 15 patients with ASD or degenerative lumbar disease who underwent corrective surgery at our hospital between January 2020 and June 2021. A 3D vascular image with bony structures was obtained by fusing results from MRI and CT. We processed the merged image and measured the distance between left and right common iliac arteries and veins at two levels: the lower end of the L5 vertebral body (Window A) and the upper end of the S1 vertebral body (Window B). Results: The mean sizes of Window A and Window B were 29.7 ± 10.7 mm and 36.9 ± 10.3 mm, respectively. The mean distance from the bifurcation to the lower end of the L5 vertebra was 23.7 ± 10.9 mm. Coronal deviation of the bifurcation was, from center to left, 12.6 ± 12.3 mm, and the distance from the center of the L5 vertebral body to the bifurcation was 0.79 ± 7.3 mm. Only one case showed a median sacral vein (6.7%). Clinically, we performed OLIF51 in 12 of the 15 cases (80%). Conclusion: Evaluating 3D lumbosacral vascular anatomy using a non-enhanced MRI and CT medical image fusion technique is very useful for OLIF51, particularly for patients in whom the use of contrast medium is contraindicated.

## 1. Introduction

Adult spinal deformity (ASD) is caused by spinal malalignment. This condition affects a large number of patients, resulting in symptoms such as severe low back pain, neurological dysfunction, reflex esophagitis, cosmetic disorders, and mental disorders [1,2]. Excellent results have been reported for ASD surgery [3,4]. Minimally invasive surgery (MIS) has been advocated to reduce surgical complications of ASD [5]. Lateral lumbar interbody fusion (LLIF) has gained popularity for the surgical treatment of ASD. Oblique lumbar interbody fusion at L5–S1 (OLIF51) is a useful technique for recreating good lumbosacral lordosis, and requires an approach from the corridor between common iliac vessels in a lateral position [6]. However, a small vascular window at the L5–S1 level is a contraindication for OLIF51 [7]. Imaging modalities, such as angiography, MRI, and contrast-enhanced computed tomography (CT), are utilized to evaluate the vascular anatomy in this lumbosacral region for clinical use [8]. Rates of vascular injury have been reported as 4.3% for OLIF51 and 3.3% for anterior lumbar interbody fusion at L5–S1 (ALIF51) [9,10].

Contrast-enhanced CT reconstruction vascular images can provide spine surgeons with multidirectional images that facilitate an understanding of the relationship between the relevant vascular structures and the L5–S1 disc. However, contrast media can cause allergy reactions, renal damage, and hypotension and require prolonged radiation exposure [11,12]. To address such problems, we report herein a novel technique to provide a three-dimensional (3D) vascular image containing bony information, created using vascular images from magnetic resonance imaging (MRI) and bone images from 3D CT.

## 2. Materials and Methods

### 2.1. Patient Population

The present study was approved by our institutional review board. We obtained fully informed consent from each patient prior to participation in this study. From January 2020 to June 2021, a total of 15 patients with ASD or degenerative lumbar disease who underwent corrective surgery at our hospital were included in the present study (Table 1, Figure 1 and Figure 2). The inclusion criteria for this study were: age ≥40 years and planned OLIF51(Sovereign spinal system^®^, Medtronic, Sofamor Danek, Minneapolis, MN, USA). We excluded young male patients who needed L5–S1 fusion because of unacceptable complications of retrograde ejaculation.

### 2.2. Image Technique

Vascular images were taken using Signa HDxt 1.5-T platform (General Electric, Boston, MA, USA). The image sequence was fast imaging employing steady-state acquisition (FIESTA) from steady-state free precession (SSFP), using the following settings: slice width, 3.0 mm; slice interval, 0 mm; echo time, 2.1 ms; flip angle, 120°; matrix, 192 × 288; accumulations, 2; bandwidth, 83.33 Hz/pixel; field of view, 360 mm. The imaging range was from the upper end of the L4 vertebra to the lower end of the S1 vertebra. The imaging cross-section bisected the angle between the upper end of the L4 vertebra and the lower end of the S1 vertebra. The number of slices was approximately 30, and the scan time was about 2 min.

The bone 3D image was created using an Aquilion lightning platform (Canon, Tokyo, Japan). The imaging conditions were: tube voltage, 120 kV; scan speed, 0.50 s; slice width, 0.5 × 80 mm; and helical pitch, 65.0. From these data, the 3D spine image was reconstructed using AIDR 3D enhanced eStrong software (Canon).

### 2.3. CT–MRI Fusion Image

The fusion image combining 3D vascular image with bony structure was obtained using Synapse Vincent version 3.3 software (Fujifilm, Tokyo, Japan). The first step was justification of the CT image (soft tissue mode) and MR image (FIESTA) in a 2D image using high signals from CSF and the vasculature. The second step was obtaining the 3D bony image from CT, setting the transparency rate of the ilium bilaterally at 0.2 to make the image of the sacrum clear under the L5 vertebral level. With MRI, images of the inferior vena cava (IVC), median sacral vessels, and common iliac vessels were made manually. The final step was merging of the bony image from CT and the vascular image from MRI (Figure 3).

### 2.4. Evaluation of Merged Images

To reduce subjective errors in calculations, each image was justified, and measurements were made by two individuals. We measured the distance between left and right common iliac arteries and veins at the lower end of the L5 vertebral body (Window A; Figure 4 ①) and at the upper end of the S1 vertebral body (Window B; Figure 4 ②). In addition, vertical distances from the bifurcations of the arteries and veins to the lower end of the L5 vertebral body were measured (Figure 4 ③④). Coronal deviation from the center of the L5 vertebral body to the vascular bifurcation was also measured. Two senior surgeons evaluated OLIF51 feasibility according to our grading system by Window A: Grade 1 (easy to perform), >20 mm; Grade 2 (possible but difficult), 15–20 mm; and Grade 3 (impossible), <15 mm.

Kappa coefficients were determined to measure inter- and intra-observer reliability.

## 3. Results

### 3.1. Radiological Evaluations

The mean sizes of Window A and Window B were 29.7 ± 10.7 mm and 36.9 ± 10.3 mm, respectively. The distances from the bifurcation of the common iliac vein and artery to the lower end of the L5 vertebral body were 23.7 ± 10.9 mm and 33.6 ± 10.1 mm, respectively. Coronal deviations of the common iliac vein and artery were 12.6 ± 12.3 mm (right side) and −0.79 ± 7.3 mm (left side), respectively (Table 2). Only one case showed a median sacral vein (8%).

### 3.2. Clinical Evaluation

OLIF51 feasibility was judged as Grade 1 in nine cases, Grade 2 in three cases, and Grade 3 in three cases. Kappa values for inter-and intra-observer reliability were 0.77 and 0.88, respectively. We changed the planned OLIF51 to transforaminal lumbar interbody fusion at L5–S1 (TLIF51) in all three Grade 3 cases, which were considered “impossible” due to an excessively narrow window in two cases, and because of the presence of a vascular anomaly in one case (Figure 5). Postoperative spinopelvic alignment was improved to normal values in all patients. Mean Oswestry disability index (ODI) improved from 76 ± 15% to 38 ± 12%, postoperatively. One patient showed a complication of relatively massive bleeding (560 mL) from the median vein. No other complications, such as neurological deficit, visceral injury, or endplate fracture were encountered.

## 4. Discussion

Lumbar interbody fusion has been used for the treatment of spinal disorders since the 1930s [13]. A constant endeavor of spine surgeons around the world has been to provide better patient outcomes, leading to continued advances in these procedures over the past century. Interbody fusion is now possible through minimally invasive techniques, offering distinct advantages such as reductions in soft tissue injury, postoperative pain, hospital stay and recovery time [14]. Recently, anterior approaches have gained favor due to the adequate access to the disc space and the large surface area for interbody fusion [15]. Restorations of lumbar lordosis and foraminal height are also favorable compared to posterior techniques [16]. OLIF came into vogue as a means of mitigating the approach-related difficulties encountered with LLIF and ALIF [14,17]. While the efficacies of this approach at the L2–5 lumbar levels are well established [18], OLIF51 seems more difficult due to the presence of major abdominal vessels overlying the disc space [19]. Controversy remains in the literature regarding the feasibility of OLIF51. If this controversy could be resolved, we may have an approach that is minimally invasive and allows anterior access to all lumbar levels in a single position with minimal complications and better outcomes.

The development of any technique needs adequate anatomical research to avoid unexpected outcomes during the early period of adoption. A study of 21 cadavers concluded that the oblique retroperitoneal corridor was present at all levels from L2 to S1 [20]. However, another study by Liu et al. concluded that OLIF51 was unsuitable in 18.3% of males and 10% of females [21]. They also recommended a thorough evaluation of imaging modalities to assess overlying vessels at this level and thus clarify the feasibility of OLIF preoperatively. The present study utilized fusion images of CT and MRI (Figure 3 and Figure 5), to the best of our knowledge representing the first description of this method. Cadaveric [20], CT angiographic [21], and MRI-based studies [19] have been reported previously. CT provides excellent visualization of the bony anatomy, while MRI is suitable for displaying soft tissue contrast and neurovascular structures [22]. No single imaging technique currently provides both these details. CT–MRI fusion is fairly new and enables creation of high-quality images to provide large amounts of information and visualize intricate anatomical details with great accuracy. In fact, the precision of such fusion images has been applied in the highly demanding surgical treatment of various skull base pathologies [23]. In our study, to reduce subjective errors in calculations, each image was justified and measurements were made by two individuals; then, mean values were considered. Fusion images also provided the surgeon with insights into the 3D anatomy of the surgical area preoperatively, thus allowing an approach with precision not provided from routine MRI and CT. Similarly, as stated by Li et al. [24], 2D assessment of the vasculature provides a limited assessment of the 3D nature of vessels, preventing judgement of whether these structures represent technically challenging obstructions. In their assessment of the operative window for OLIF51 surgery, Liu et al. [18] utilized CT angiography with 3D reconstruction images. We believe that our imaging technique avoids the dye-related side effects that may arise during angiography procedures [25]. CT–MRI fusion techniques have opened new treatment options and have recently been utilized to allow the treatment of spinal pathologies in combination with navigation techniques [22], and this option seems applicable to OLIF surgery in the future.

In our study, operative windows at two levels (the lower end plate of L5 and upper end plate of S1) were calculated in consideration of the fact that the OLIF cage is rectangular and the measurement at the lower border of L5 is more important. Mean distances between iliac vessels were 29.7 ± 10.7 mm and 36.9 ± 10.3 mm, respectively, considerably different from values reported in previous anatomical studies of 14.8 ± 6.9 mm [20], 10 ± 8.3 mm [19], and 15.9 ± 9.3 mm [21], but similar to those documented by Choi et al. (27 ± 9.4 mm) [8] and Song et al. (29.46 mm) [26]. We believe that the distance between iliac vessels is surgically more practical than the distance between the mid-sagittal plane and left iliac vessels [8]. Senior surgeons decided to grade feasibility using Window A Grade 1—> 20 mm (easy to perform), Grade 2—15–20 mm (possible but difficult), and Grade 3—<15 mm (impossible). This was a modification of the recommendations by Choi et al. [8], which suggested that the corridor is inadequate in patients with values <15 mm with absence of perivascular fat, rather than the earlier recommendations of <1 cm based on the dimensions of the LLIF cage.

Using the measurements obtained, two of our 15 cases were classified as Grade 3 based on the narrow vascular window, so OLIF surgery was deferred. Another patient showing a vascular anomaly underwent TLIF51 instead. Three of the 15 patients (20%) thus had an inadequate corridor for OLIF51 in our study. Only one complication related to bleeding from median sacral vessels was encountered in our study. Molinares et al. [19] found that the corridor was inadequate in 31% cases based on MRI of 133 individuals. Similarly, Liu et al. [21] in their CT angiography study reported OLIF51 was not possible in 28.3% of cases. We believe that differences in grading criteria and measurements as mentioned above might have contributed to such discrepancies, but our technique providing better imaging might also have enabled OLIF in up to an additional 10% of patients, thus broadening the indications for this technique. Additionally, patients in our study were actually operated upon, confirming the practical feasibility of this method.

Certain limitations of this study should be kept in mind. First, the sample size was small. Second, the results of this study combined with clinical intraoperative measurements would increase the strength of the study, since vascular positions might change in the lateral position. Since no such accurate measuring technique is currently available, we were unable to obtain such measurements. We believe integration of navigation systems might open such avenues in the future.

## 5. Conclusions

Clarification of the lumbosacral vascular anatomy using 3D non-enhanced MRI and CT medical image fusion techniques appears very useful for evaluating the feasibility of OLIF51, particularly for patients in whom use of contrast media is contraindicated.

## Figures and Tables

**Figure 1 diagnostics-11-01744-f001:**
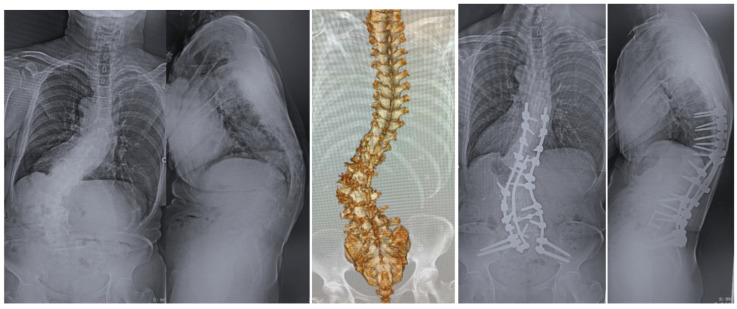
72 year old woman, adult spinal deformity, OLIF from L1 to S1 and posterior fusion.

**Figure 2 diagnostics-11-01744-f002:**
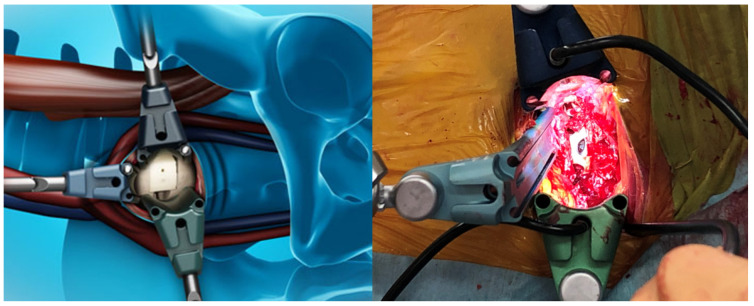
OLIF 51.

**Figure 3 diagnostics-11-01744-f003:**
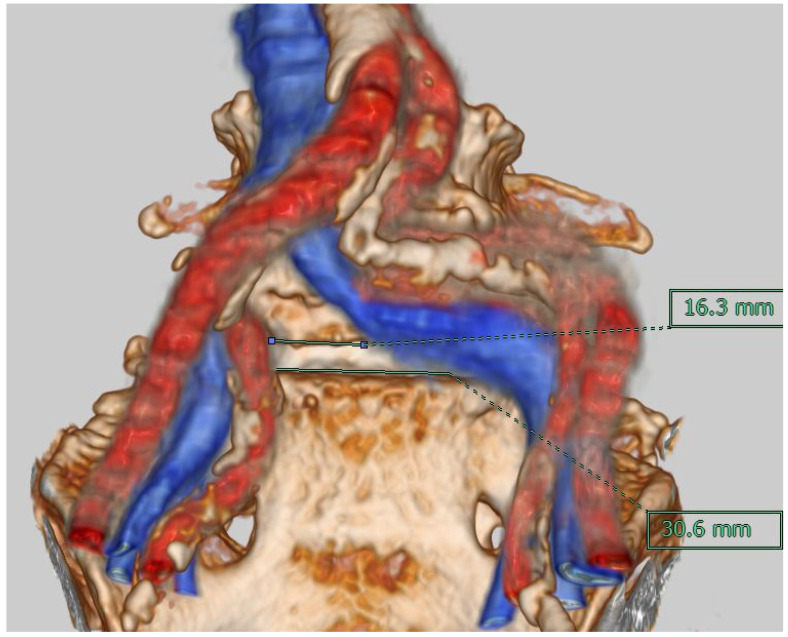
MRI and CT medical image fusion.

**Figure 4 diagnostics-11-01744-f004:**
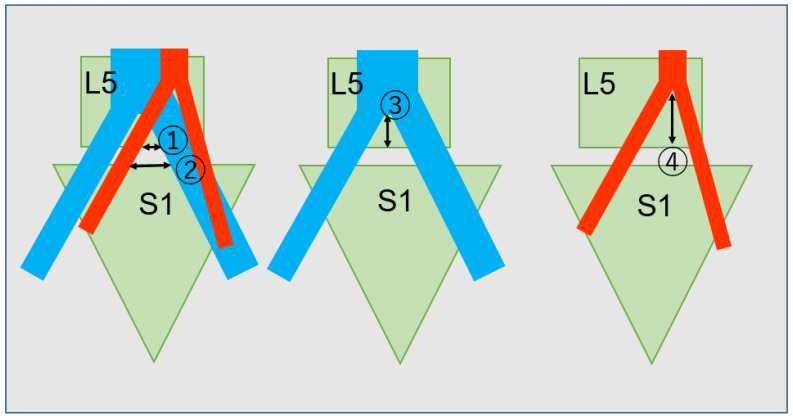
Evaluation of the merged image. ① Window A; ② Window B; location of bifurcation of artery ③; and vein ④. Blue and red indicate vein and artery, respectively. L5 and S1 mean L5 vertebra and S1 vertebra, respectively.

**Figure 5 diagnostics-11-01744-f005:**
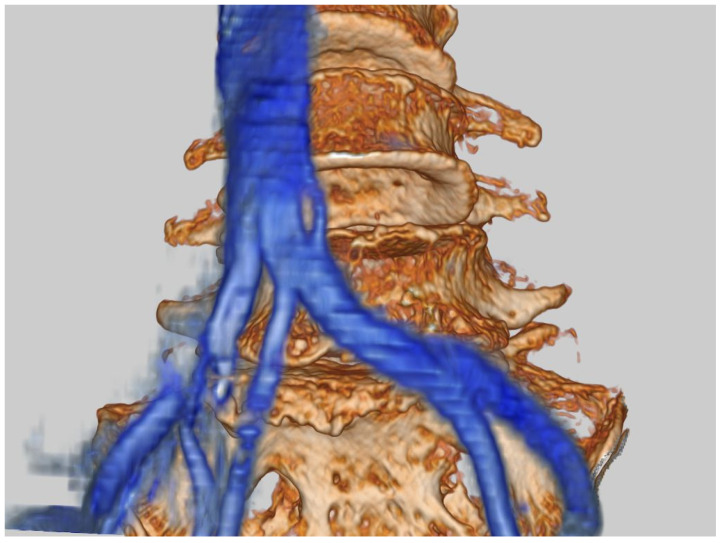
Anomaly of veins and a narrow window case.

**Table 1 diagnostics-11-01744-t001:** Patient demographics.

	*N* = 15
Gender (Man: Woman)	0:15
Age (mean ± S.D.) (year)	75.3±8.6
Height (mean ± S.D.) (cm)	154.2 ± 8.2
Body weight (mean ± S.D.) (kg)	59.3 ± 11.8
Body mass index (mean ± S.D.) (kg/m^2^)	24.9 ± 4.4
History of abdominal surgery	Uterus surgery 2

**Table 2 diagnostics-11-01744-t002:** Radiological evaluation.

Radiological Evaluation	*N* = 15
Inter common iliac vessel distance at L5 caudal endplate (mm)	29.7 ± 10.7
Inter common iliac vessel distance at S1 cranial endplate (mm)	36.9 ± 10.3
Distance from bifurcation of common iliac vein to L5 caudal endplate (mm)	23.7 ± 10.9
Distance from bifurcation of common iliac artery to L5 caudal endplate (mm)	33.6 ± 10.1
L5 vertebral body height (mm)	24.8 ± 3.5
Coronal deviation of bifurcation of common iliac vein (mm) (+;right, −;left)	12.6 ± 12.3
Coronal deviation of bifurcation of common iliac artery (mm) (+;right, −;left)	−0.79 ± 7.3

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
