# Peer review of "Assessment of 3D Lumbosacral Vascular Anatomy for OLIF51 by Non-Enhanced MRI and CT Medical Image Fusion Technique"

_diagnostics, 2021, doi:10.3390/diagnostics11101744_

Round 1
Reviewer 1 Report
In the study an CT-MRI fusion images for assesing the morphometric parameters of the iliac vessels in the region of vertebra L5 and S1 is investigated. The technique is new and has a major advantage in that is does not require intravenous contrast.
The authors report morphometric parameters and the size of the vascular windows at the L5 - S1 level. These are important when planning OLIF surgery as a narrrow vascular window or low lying aortic bifurcation may preclude complicate or preclude this procedure.
The authors introduce a grading system for the size of the vascular window. However, they do not adequately explain (in my opinion) which measurements are used fro this grading system. Grade 1 is easy to perform, grade 2 is possible but difficult and grade 3 is impossible. Three cases were classified as grade three and these patients instead underwent TLIF. Outcomes of the all procedures were sucessful (although follow-up is not reported).
My main suggestion is to include a control group who underwent classic CT angiography to compare the outcomes.
Also an English revision is required. There are several typos and some parts should be phrased better (e.g. please explain the grading system in the Methods in more detail).
Author Response
The authors introduce a grading system for the size of the vascular window. However, they do not adequately explain (in my opinion) which measurements are used for this grading system.
Thank you for your important comment. For grading system, we use window A; which is the distance between left and right common iliac artery and vein at lower end of L5 vertebral body.
According to your advice, we changed the sentence as follows in Materials and Methods part.
Two senor surgeons evaluated OLIF 51 feasibility according to our grading system using window A; grade 1- > 20 mm (easy to perform), grade 2-15-20 mm (possible but difficult) and grade 3- <20 mm (impossible).
My main suggestion is to include a control group who underwent classic CT angiography to compare the outcomes.
We appreciate your valuable comment. As you know, CT angiography needs contrast medium. Unfortunately, before this new imaging technique, we evaluate L5-S1 OLIF accessibility only by MRI. In the next paper, we’d like to perform comparative study.
Also an English revision is required.
We are sorry about that. We sent this manuscript to English grammar check.
Reviewer 2 Report
The authors report a novel technique when fusing bony imaging on CT with vascular imaging on MRI to assess for the Safety of the OLIF 51 procedure.
The authors describe their technique well in a manner that is easy for surgeons who have access to this technology to reproduce. They also attempt to develop a grading system on the feasibility of the OLIF 51 procedure.
I think this manuscript is worthy of publication with some moderate changes and editing of the English language (there are several typos, as well as grammatical and syntax errors).
I would like more information of the inter- and intra-observer reliability with regard to the measurements for each patient. Further, I would recommend grading these images with more than 2 surgeons to give further validity to the grading system.
If these measurements reveal a relatively high inter- and intra-observer reliability, then I think this is one of a handful of useful techniques to evaluate for the safety of the OLIF 51 procedure, especially at institutions without access to navigation.
Author Response
I think this manuscript is worthy of publication with some moderate changes and editing of the English language.
We are sorry about that. We sent this manuscript to English grammar check.
I would like more information of the inter- and intra-observer reliability with regard to the measurements for each patient. Further, I would recommend grading these images with more than 2 surgeons to give further validity to the grading system.
We’ d like to thank you for the important comments. We performed Kappa coefficient test. The inter-and intra-observer reliability (Kappa value) were 0.77 and 0.88, respectively.
If these measurements reveal a relatively high inter- and intra-observer reliability, then I think this is one of a handful of useful techniques to evaluate for the safety of the OLIF 51 procedure, especially at institutions without access to navigation.
Thank you for your important comment. We are encouraged with your wonderful comment. Actually, we have the O-arm and Stealth station navigation system Spine 7R and perform OLIF 51 with this navigation.
